# Contrast-Enhanced Ultrasonography in the Diagnosis and Treatment Modulation of Breast Cancer

**DOI:** 10.3390/jpm11020081

**Published:** 2021-01-30

**Authors:** Ioana Boca (Bene), Sorin M. Dudea, Anca I. Ciurea

**Affiliations:** Department of Radiology, “Iuliu Hatieganu” University of Medicine and Pharmacy, 400000 Cluj-Napoca, Romania; sdudea1@gmail.com (S.M.D.); ancaciurea@hotmail.com (A.I.C.)

**Keywords:** contrast-enhanced ultrasound, breast cancer, SonoVue, neoadjuvant chemotheraphy, sentinel lymph node

## Abstract

The aim of this paper is to highlight the role of contrast-enhanced ultrasound in breast cancer in terms of diagnosis, staging and follow-up of the post-treatment response. Contrast-enhanced ultrasound (CEUS) is successfully used to diagnose multiple pathologies and has also clinical relevance in breast cancer. CEUS has high accuracy in differentiating benign from malignant lesions by analyzing the enhancement characteristics and calculating the time-intensity curve’s quantitative parameters. It also has a significant role in axillary staging, especially when the lymph nodes are not suspicious on clinical examination and have a normal appearance on gray-scale ultrasound. The most significant clinical impact consists of predicting the response to neoadjuvant chemotherapy, which offers the possibility of adjusting the therapy by dynamically evaluating the patient. CEUS is a high-performance, feasible, non-irradiating, accessible, easy-to-implement imaging method and has proven to be a valuable addition to breast ultrasound.

## 1. Introduction

Contrast-enhanced ultrasound (CEUS) in breast pathology is used for various purposes, from differentiating benign from malignant lesions and evaluating the extent of the disease to assessing the response to neoadjuvant chemotherapy (NAC) [1,2,3]. The purpose of this paper is to highlight the role of CEUS in breast cancer diagnosis, staging and follow-up of the post-treatment response.

Although CEUS is a non-irradiating, non-invasive and safe imaging technique, it involves contrast administration, so patient preparation is essential and venous access is required.

(a) Contrast media

The purpose of contrast administration in breast ultrasound is to characterize breast lesions, visualize and identify the sentinel lymphatic channels (SLCs) and the sentinel lymph node (SLN). To assess breast lesions, the contrast is administered intravenously using a 20gauge cannula while for the visualization of the SLCs and SLN it is injected subcutaneous in the periareolar region [4,5].

CEUS uses two types of contrast agents: SonoVue^®^ ((Bracco Spa, Milan, Italy) and Sonazoid^®^ (Daiichi Sankyo Corporation, Tokyo, Japan).

SonoVue^®^ is sulfur hexafluoride and is the most widely used contrast agent consisting of microbubbles with a mean diameter of about 2.5 μm (90% having a diameter less than 6μm). It is important to exclude contraindications such as hypersensitivity to sulfur hexafluoride (or any of the components of SonoVue^®^), recent acute coronary syndrome or clinically unstable ischemic cardiac disease, acute cardiac failure, severe rhythm disorders, known right-to-left shunts, severe pulmonary hypertension (pulmonary artery pressure >90 mmHg), uncontrolled systemic hypertension and respiratory distress syndrome [6].

Sonazoid^®^ is perfluorobutane within a hydrogenated egg phosphatidylserine shell, which contains approximately 8 μL microspheres/mL with a median diameter of approximately 2.6 μm. Contraindications are allergies to eggs or egg products (hydrogenated egg phosphatidylserine sodium in Sonazoid may cause allergic symptoms), hypersensitivity to other components of Sonazoid, recent acute coronary syndrome or clinically unstable ischemic cardiac disease, adult respiratory distress syndrome, severe emphysema, pulmonary vasculitis, or history of pulmonary emboli, known right-to-left shunt, severe pulmonary hypertension, or uncontrolled systemic hypertension [7].

(b) CEUS examination protocol

Assessment of breast lesions. Before the contrast examination, the lesion of interest is assessed on both gray-scale and color Doppler ultrasound to identify the scan plane that includes the largest diameter of the lesion with the richest vascularization. It is important also to avoid macrocalcifications and areas with acoustic attenuation. At present, there is no standard protocol for the evaluation of breast lesions. In most studies, the examination involved the use of high-frequency linear-array transducers (usually L_9-3_ type transducer), the setting of the mechanical index to 0.06–0.08, gain of 100–120 dB, single focus and image depth of approx. 3–4 cm.

SonoVue^®^ is administered as a bolus of 2.4–4.8 mL followed by a flush of 5–10 mL sodium chloride 0.9%. During the examination, the scanning plane should remain unchanged and the examiner should apply no pressure to the transducer. To avoid artifacts, the patient must stay still and avoid extensive breathing movements, while images are recorded for at least 180 s [8,9,10,11,12].

Sonazoid^®^ is administered as a bolus of 0.5 mL, followed by 10 mL standard saline flush [13].

After contrast administration the breast lesion is assessed in terms of the degree of uptake, internal homogeneity, margins, presence of perfusion defects, penetrating vessels or perilesional uptake. To determine the hemodynamic indicators, the examiner draws a region of interest (ROI, usually the most perfused area of the lesion of about 16 mm^2^), where the time-intensity curve (TIC) of the contrast transit (Figure 1) is recorded and by using an analysis software, the quantitative parameters are computed [14].

The CEUS protocol for detecting SLNs is different from the one used to evaluate breast lesions. The contrast agent is injected subcutaneously in the areolar region of the upper-outer quadrant. The LCs are visualized and seconds later (depending on the type of contrast) the SLNs.

SonoVue^®^, injected in amount of 0.2–0.5 mL, has a transition time of around 15–45 s from injection to its appearance in the SLN where it stays for 1 to 3 min [5].

Sonazoid^®^, injected in amount of 2 mL, has an average arrival time in the SLN of approximately 5.3 min and the contrast enhancement does not decrease even over 10 min [15].

Thus, this longer transition time in the case of Sonazoid^®^ could be an advantage over SonoVue^®^, which, due to the fast wash-out, leaves SLN very quickly and may mislead radiologists about the correct SLN. In both cases, because the microbubbles have a larger diameter than the endothelial gaps, post-injection massage for 10–30 s could be useful. Perhaps a contrast agent with smaller microbubbles would prove to be more effective and increase diagnostic accuracy [16].

(c) Technical aspects

The reflected signal intensity depends on the concentration of microbubbles and the frequency of the ultrasound beam. In “in vitro” studies, most of the echogenicity is provided by bubbles with a corresponding resonance frequency of around 3 MHz [17]. This frequency is characteristic for convex-array transducers used in the evaluation of abdominal organs. Breast ultrasound uses linear-array transducers with high frequency; therefore, breast lesions’ CEUS diagnosis could be altered. Given this perspective, Wang et al. [18] conducted a study, including 51 breast lesions in which they compared the effectiveness of transducers with different frequencies (C_5—1_ versus L_12—5_) using SonoVue^®^ as a contrast agent and they observed no difference in the use of the two transducers regarding benign lesions. In malignant lesions, the low-frequency transducer was more sensitive in detecting malignant features such as perfusion defects (60.5% vs. 31.6%) and surrounding vessels (68.4% vs. 31.6%). Thus a new contrast agent with a high resonance frequency could improve breast CEUS.

Tracing the ROI is an important aspect, primarily since a standardized mode for its determination has not yet been described. Some authors placed a single ROI in the area of greatest enhancement while others included the entire tumor and its margins [19,20]. Others analyzed the quantitative parameters in different ROIs (ROI 1—the area with the greatest enhancement, ROI 2—the area with hyperenhancement avoiding necrotic regions, ROI 3—the entire tumor avoiding the surrounding parenchyma, ROI 4 the normal-appearing parenchyma from the same acquisition plane). The authors stated that ROI 1 was a subjective measurement. At the same time, ROI 2 and ROI 3 were objective measurements, eliminating interobserver variability, but they represent the average values of contrast uptake, causing misdiagnosis. Individual particularities of the patients (heart rate, body weight) could influence the kinetics of CEUS. The perfusion values for ROI 1, ROI 2, and ROI 3 were divided by normal parenchymal values of each patient (ROI 4) to obtain adjusted values [21].

## 2. Evaluation of Breast Lesions with CEUS

At CEUS, breast lesions are differentiated by analyzing qualitative characteristics and measuring quantitative parameters (Table 1) [14].

The appreciation of tumor angiogenesis helps diagnose malignant lesions, evaluate treatment response and establish the prognosis. Doppler ultrasound detects large vessels (>100–200 µm), while CEUS can observe tumor microcirculation by using contrast agents that are approximately 2.5 µm in size. The characteristics of perfusion in malignant tumors are high velocity and blood perfusion due to vessels with an irregular caliber and course, the presence of arterio-venous shunts and sinusoids and the absence of the muscular layer in the newly-formed capillaries [9,17].

CEUS can differentiate between benign and malignant lesions by analyzing the acquired images. Benign characteristics on CEUS include sharp margins, no enhancement or homogeneous centrifugal enhancement of the lesion and dendritic branching vessels. In contrast, malignant lesions have blurred margins, a heterogeneous centripetal enhancement, with filling defects and distorted vessels [1,10,14,17,22,23]. Other criteria suggestive of malignancy are early uptake (intense wash-in and wash-out) due to arteriovenous shunts and the presence of peripheral enhancement due to the higher density of microvessels in the peritumoral areas [24], an aspect which corresponds to the histopathological reports. Regarding quantitative parameters, malignant lesions have a significantly shorter time to peak, high peak intensity and increased wash-in slopes [9,14].

CEUS quantitative parameters obtained from time-intensity curves (peak, TTP, RBV, and RBF) and optimal cut-off points prove to be useful for distinguishing between benign and malignant lesions (Table 2) [1].

## 3. Correlation between Prognostic Factors and CEUS Characteristics and Quantitative Parameters

The possibility of predicting the molecular subtype with the help of CEUS is an important aspect in clinical practice and it is also the subject of innovative radiomic studies in senology [25].

In terms of hormone receptors, breast tumors with negative estrogen receptors (ER) present a high-intensity peak, ill-defined margins, centripetal enhancement, vascular perfusion defects, fibrosis and central necrosis. In positive ER tumors, the necrotic tissue is less represented due to a lower cell proliferation rate and they have a better prognosis [9,14,26].

Ki-67 is an independent prognostic factor as tumors with a positive Ki-67 are tumors with a poor prognosis due to high proliferative activity. These tumors show perilesional and heterogeneous enhancement and perfusion defects, which may be due to rapid tumor growth, hypoxia, necrosis and fibrosis. Regarding the quantitative parameters, when compared with negative Ki-67 lesions, the tumors with positive Ki-67 have a fast wash-in and a slower wash-out velocity and a high maximum intensity of the curve [9,14,17,26,27,28].

Her2 is a protooncogene on chromosome 17q. Its overexpression is associated with a lesion’s malignant transformation, rapid progression and occurrence of metastases and a poor prognosis. Tumors with positive Her2 present high enhancement and perfusion defects (Figure 2) [9,17,26,29,30].

Vraka et al. [14] found an association between CEUS quantitative parameters obtained from time-intensity curves (peak, TTP, RBV, and RBF) and pathological prognostic factors (Table 3).

Breast tumors with a high histological grade have blurred margins due to an ongoing angiogenetic process, which corresponds to the proliferative activity, are more susceptible to necrosis, present penetrating vessels and perfusion defects. Hence, the contrast takes longer to fill the tumor [9,14,29,31,32,33]. The enhancement is heterogeneous and perilesional due to a higher peripheral/central ratio of microvessel density, compared to tumors with a lower histological grade. There is also a difference regarding the rise time (RT) and TTP (Table 4) [14,26,34].

The area of malignant lesions is significantly higher in CEUS than in gray-scale ultrasound, especially in tumors with positive ER, due to angiogenesis and a balanced spatial distribution of tumor blood vessels. Histopathological examinations found that 85% of the invasive carcinomas had ductal carcinoma in situ (DCIS) at the periphery, which, without microcalcifications or dilated ducts, is unidentifiable on gray-scale ultrasound. There are also benign lesions that can increase the uptake area, such as inflammatory processes or hypervascular adenosis surrounding malignant lesions [2,10,12].

## 4. Evaluation of Breast Lesions: The Added Value of CEUS to Ultrasound and Comparison with MRI

The association of CEUS with gray-scale ultrasound yields a superior diagnostic performance (Table 5), especially for BIRADS 3–5 lesions. The incorporation of CEUS into the BIRADS system was effective by using the 5-point score [4,10,35,36]:Score 1 = BIRADS 3: no enhancement, clear delimitation of the lesion from the adjacent parenchyma;Score 2 = BIRADS 4A: synchronous and iso-enhancement with the adjacent parenchyma, without clear delimitation;Score 3 = BIRADS 4B: early, homogeneous/heterogeneous enhancement, regular shape (round/oval), sharp margins, size equal to/smaller than at gray-scale ultrasound;Score 4 = BIRADS 4C: early heterogeneous enhancement, lesion size larger than at gray-scale ultrasound, irregular shape, ± perfusion defects, ± crab-claw-like enhancement;Score 5 = BIRADS 5: early, heterogeneous and typical crab-claw-like enhancement, irregular margins, lesion size larger than at gray-scale ultrasound, ± perfusion defects.

There is also a good agreement between US + CEUS and MRI regarding differentiation of benign from malignant lesions [1]. Both imaging techniques have an important role in terms of characterization of lesions <1 cm, as well as in the assessment of BIRADS 4a lesions by reclassifying (some of them) in BIRADS 3, thus avoiding unnecessary biopsies and reducing patient anxiety. MRI is an extremely sensitive method in detecting and evaluating breast lesions and also in detecting supplementary lesions, which usually require a second-look ultrasound. This technique can increase the detection rate to 57.5% for additional malignant lesions [36,37]. CEUS proves to be a feasible method in detecting supplementary lesions that are occult at mammography, ultrasound, second-look ultrasound, but visible on MRI, lesions that will change the surgical plan. In these situations, CEUS guided biopsy becomes an option because there is no need to recall and schedule the patient for MRI guided biopsy, which is a difficult, time-consuming procedure and sometimes the lesions are not approachable for biopsy [2,38].

## 5. Usefulness of CEUS

The diagnostic accuracy of CEUS is higher in patients over 35 years, in the case of lesions with a diameter above 20 mm and in cases where the distance from the lesion’s deep edge to the pectoralis major muscle is equal or less than 3.05 mm [39].

In addition to the above mentioned benefits or indications of CEUS: the possibility of differentiating between benign and malignant lesions, the possibility of predicting the molecular subtype, detection and characterization of LCs, SLN and predicting the response to neoadjuvant chemotherapy, CEUS has some notable advantages. Compared to MRI, CEUS is an accessible imaging technique, the examination time is shorter, the patient are examined in a comfortable position and it can be performed on patients with ferromagnetic metal implants, pacemakers, claustrophobia or with contraindication to Gadolinium administration.

CEUS is a useful imaging method in assessing the success rate of microwave ablation (MVA) for benign breast lesions. Intralesional and perilesional enhancement assessment before and one hour after MVA at CEUS and MRI are useful in assessing the effectiveness of ablation, with a success rate of 87.32% for CEUS and 82.93% for MRI [40].

Yao et al. [41] went further in evaluating the usefulness of CEUS and SonoVue^®^. The authors analyzed in “in vivo” and “ex vivo” animal models studies the improvement of the therapeutic efficacy of high intensity focused ultrasound (HIFU) thermal ablation of breast tumors by using SonoVue^®^ microbubbles. The microbubbles present at the level of the tumor act as cavitation nuclei and enhance tissue heating during HIFU treatment producing coagulation necrosis.

## 6. Diagnosis Limitations of CEUS

CEUS, like any imaging technique, has its limitations. It has a low diagnostic performance in detecting DCIS, early-stage invasive ductal carcinoma (IDC) and also regarding rare and unusual types of breast cancer. For instance, triple-negative cancers in CEUS have a regular shape, sharp margins, the same size as in gray-scale ultrasound, present hypo or iso-enhancement with a slow wash-in and no penetrating vessels or crab-claw pattern (Figure 3). Mucinous carcinomas have the same size or are smaller compared with gray-scale ultrasound. They can present either a slow wash-in with hypo-enhancement due to the lack of vascularization and rich mucus content or can have a fast wash-in, with intense and heterogeneous uptake of the contrast agent [8]. Medullary carcinomas have a regular shape, well-defined margins, homogenous enhancement and no change in size before and after contrast enhancement, but they can present penetrating or tortuous surrounding vessels [42].

To exceed these CEUS limits and improve diagnostic performance, computer-aided design systems could be used as they have been shown to be effective in mammography techniques [43,44].

## 7. CEUS in the Staging and Treatment of Breast Cancer

### 7.1. CEUS in Sentinel Lymph Node Assessment

Sentinel lymph nodes (SLNs) represent the first lymph nodes to receive lymphatic drainage from breast carcinomas and indicate positive or negative axilla [45]. Their status is an important prognostic factor in patient survival, disease recurrence and influences the patient’s therapeutic management, especially regarding the administration of neoadjuvant chemotherapy (NAC). Given these aspects, axillary dissection is currently reserved only for cases in which SLNs show micro- or macrometastases; otherwise, it is replaced by sentinel node biopsy (SLNB) [5]. Predictors of lymph node metastases at CEUS are tumor area ≥5.37 cm^2^, tumor width ≥2.35 cm and tumor depth ≥1.95 cm [11].

SLN detection uses radioisotope, blue dye or both with a false negative rate of 6% [46], explained by discontinuous LCs due to tumor emboli presence [47]. Disadvantages of the radioisotope are a short half-life, a trained staff’s necessity, abiding to special legislative requirements and irradiation. At the same time, blue dye may cause skin necrosis and allergic reactions [48]. The sensitivity of CEUS in detecting SLNs is similar to that of the radioisotope (96%) and slightly lower than blue dye (100%). Therefore, this method may be considered an alternative, mainly because it has low costs and can be quickly adopted both in developed and developing countries [16,45]. CEUS increases the detection rate, especially in lymph nodes with a deep localization or in those with a prominent fatty hilum and a thin cortex, difficult to differentiate from axillary fat. CEUS is also a suitable method for more accurately differentiating benign from malignant lymph nodes and reducing unnecessary SLNB by 50% [5,49].

SLNs enhancing patterns may be useful in making the differential diagnosis. Zhao et al. and Xie et al. described three types of enhancement:type I—homogeneous enhancement.type II—heterogeneous uptake (regional or diffuse), with hypoperfusion or unperfused areas.type III—poor or no enhancement [50,51].

Type I was considered benign, types II and III were deemed malignant (Figure 4), yielding a sensitivity (Se) of 100%, a specificity (Sp) of 52%, with a positive predictive value of 64.9%, negative predictive value of 100% and a diagnostic accuracy of 64.9%. Therefore, the homogeneous enhancement has the highest negative predictive value, representing an indicator of benignity. In contrast, inhomogeneous or no enhancement is due to perfusion defects and is significantly associated with lymph node tumor infiltration [50,52]. The sentinel node’s heterogeneous appearance was observed in the case of Her2 positive tumors and in tumors ≥2 cm [9,29,33].

The use of CEUS in SLN assessment after NAC administration is reliable, but is not generally accepted. The detection rate of SLNs before NAC is 99.1% and decreases to 80.1% in the initially positive nodes after NAC. This can be explained by the fact that chemotherapy produces fibrotic changes in LCs, which makes lymph nodes challenging to detect [53,54].

A meta-analysis that included 13 articles and 876 breast cancer patients reached a pooled Se of 0.80 (95% CI: 0.76–0.84) and a pooled Sp of 0.94 (95% CI: 0.91 to 0.96) regarding the detection and the characterization of SLNs [55].

### 7.2. CEUS in Sentinel Lymphatic Channels Assessment

There are three types of sentinel lymphatic channels, thus diverse lymphatic drainage patterns to the axilla, which could also impact the accuracy of SLNB:superficial sentinel lymphatic channels (SSLCs)—which start from the subareolar lymphatic plexus and pass into the subcutaneous adipose tissue.penetrating sentinel lymphatic channels (PSLCs)—which also originate from the subareolar lymphatic plexus and pass into the breast glandular parenchyma.deep sentinel lymphatic channels (DSLCs)—which start from the breast parenchyma and pass through the breast parenchyma or into the retro mammary cellular space [56].

Detection of LCs, drainage patterns and SLNs can be performed with the help of 3D-CEUS, a method that has proven to be extremely useful, mainly because it shows the spatial anatomical relationship between these structures and guides the surgeon during the operation [57].

### 7.3. CEUS in Predicting Pathological Response After NAC

Due to the heterogeneity of breast cancers in terms of genotype, morphology and vascularization, their contrast uptake at CEUS and chemotherapy response are variable [58]. The response may be evaluated through imaging methods and it has clinical significance because in non-responders the treatment can be replaced with alternative drugs [59,60,61].

The pathological response to NAC is assessed using the Miller Payne score, based on malignant cell changes by comparing biopsied samples (before NAC) with surgical specimens (after NAC completion) [62].

The fact that CEUS is an imaging technique with high diagnostic performance in the evaluation of the breast cancer response to NAC was proven in a meta-analysis that included nine studies and 421 patients reaching a pooled Se of 0.87 (95% CI: 0.81–0.92) and a Sp of 0.84 (95% CI: 0.74–0.91) [63].

One cannot differentiate tumors with necrosis and fibrous scars from tumor residue through gray-scale ultrasound alone. The quantitative analysis of blood perfusion at CEUS (peak, peak%, TTP%) is superior to the measurement of tumor diameter regarding the prediction of complete pathological response (pCR), having a Se of 81.2% vs. 93.8% and a Sp of 94.3%vs. 54.3% while by their association Se and Sp are 93.7% and 80% respectively. Tumor diameter change is an independent predictive factor. Still, it does not occur immediately after NAC administration. However, it was found that tumor size measured by CEUS after NAC was larger than it appeared on gray-scale ultrasound and had good agreement with tumor size on surgical specimens. Peak intensity and ascending slope are lower in patients with pCR, CEUS having a Se and accuracy equivalent or even higher than MRI. Changes in these parameters correspond to the disappearance of tumor cells, degeneration and regression of tumor vessels; therefore, CEUS can be used as an alternative to MRI, especially in patients with contraindications to MRI examination [64,65,66].

Some authors evaluated NAC’s efficacy after the first cycle of chemotherapy or after two chemotherapy cycles, or even at the end of treatment. After the first cycle of chemotherapy, no significant difference in quantitative parameters was observed between responders and non-responders. Changes in tumor size occur only after the second NAC cycle [19,20,21,59,60,61].

Ultrasound elastography showed a considerably reduced strain ratio after NAC in responders compared to non-responders (2.11 ± 0.52 versus 3.71 ± 1.29) and without a significant difference at baseline but, when combined with CEUS, the Se, Sp, positive predictive value (PPV), negative predictive value (NPV) and accuracy were higher than those of CEUS or elastography alone [67].

## 8. Conclusions

CEUS is an imaging technique that assesses qualitative characteristics and quantitative parameters of breast lesions. It is a feasible and efficient technique in detecting lymphatic channels and sentinel lymph nodes. It predicts complete pathological response with increased accuracy by evaluating changes following neoadjuvant chemotherapy administration, making it extremely valuable in breast imaging.

## Figures and Tables

**Figure 1 jpm-11-00081-f001:**
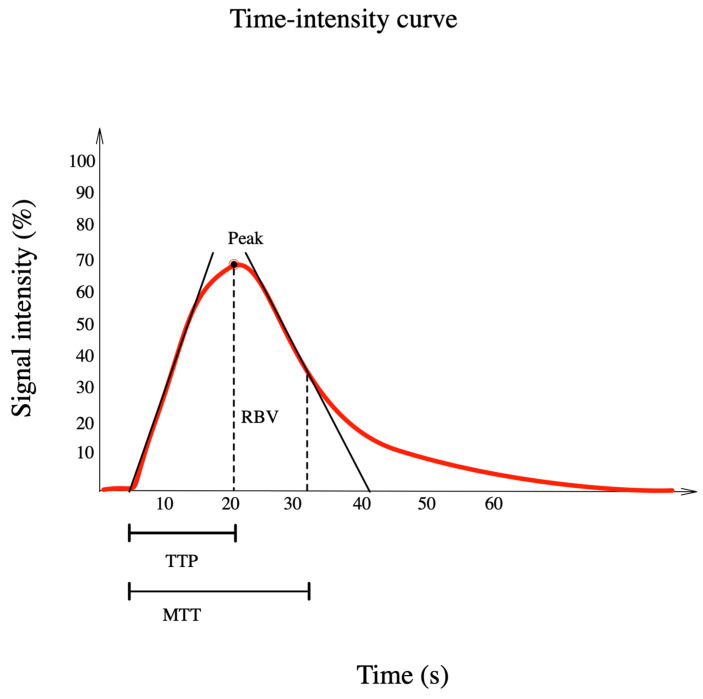
Representation of the time-intensity curve including parameters: TTP (time to peak) (s), MTT (mean transit time) (s) and RBV (regional blood volume) (mL).

**Figure 2 jpm-11-00081-f002:**
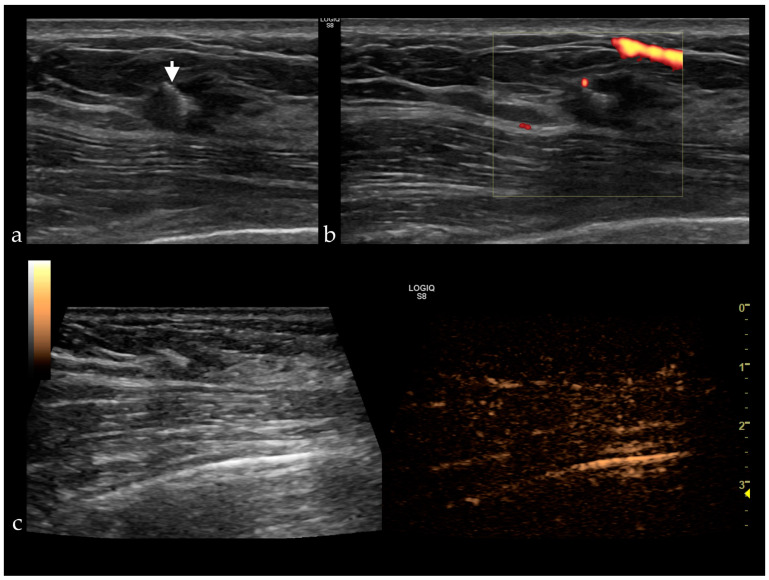
(**a**) Gray-scale ultrasound reveals a HER2 positive breast cancer presenting as a hypoechoic, non-circumscribed lesion with a clip placed in the center (white arrow), without vascularization (**b**) and no enhancement after contrast administration (**c**).

**Figure 3 jpm-11-00081-f003:**
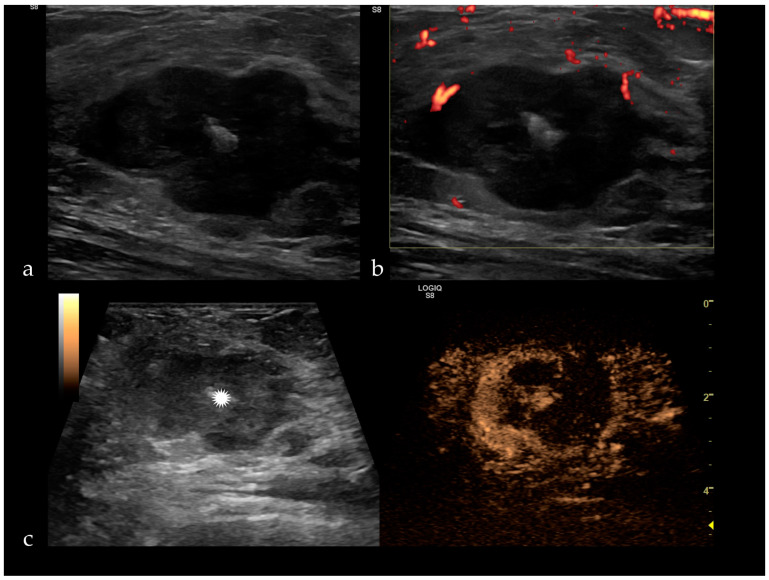
(**a**) Triple negative breast cancer appears as a hypoechoic, irregular mass, with a central hyperechogenity represented by a clip (white star), peripheral vascularization on Doppler ultrasound (**b**), inhomogeneous enhancement and filling defects after SonoVue administration (**c**).

**Figure 4 jpm-11-00081-f004:**
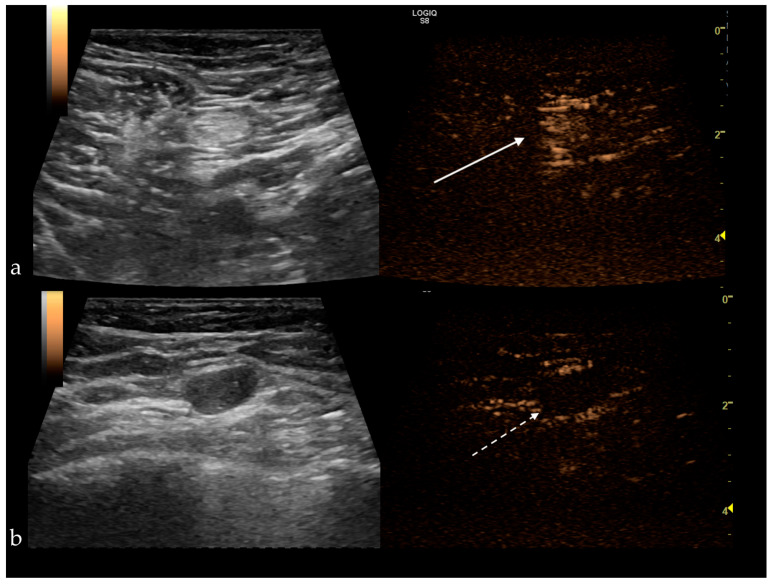
(**a**) Benign sentinel lymph node presenting a homogenous central enhancement (white arrow); (**b**) Metastatic sentinel lymph node with no enhancement after subcutaneous administration of the contrast agent (white dotted arrow).

**Table 1 jpm-11-00081-t001:** Qualitative characteristics and quantitative parameters of breast lesions at CEUS [1,4].

Breast Lesions at CEUS	
Qualitative Characteristics	Quantitative Parameters
enhancement degree compared with the surrounding breast tissue (hypo-, iso-, hyperenhancement)	Peak (%): the maximum intensity of the enhancing curve during the bolus given by the formula [(post-contrast signal−pre-contrast signal)/pre-contrast signal] × 100%
internal homogeneity (homogeneous/inhomogeneous)	TTP (time to peak) (s): the time from the appearance of the first microbubbles in the lesion to its maximum peak intensity
presence/absence of perfusion defects	MTT (mean transit time) (s): the time interval from the appearance of the first microbubbles in the lesion to the time when the peak intensity fells to half.
uptake pattern (centripetal/centrifugal)	RBV (regional blood volume) (mL): the area under time-intensity curve, reflecting total volume of contrast medium (or blood) traversing the region of interest
lesion margins (well/ill-defined >50% of circumference)	RBF (regional blood flow) (mL/sec): the relative blood flow in the selected lesion’s area, calculated as RBV/MTT
presence/absence of perilesional enhancement	

**Table 2 jpm-11-00081-t002:** CEUS quantitative parameters and cut-off points for distinguishing between benign and malignant lesions.

	Peak (%)	TTP (s)	MTT (s)	RBV (mL)	RBF (mL/s)
Benign	23.18 ± 12.4	34.79 ± 11.60	47.54 ± 13.94	1277.25 ± 14.98	26.17 ± 14.98
Malign	29.22 ± 6.78	25.92 ± 11.09	47.45 ± 15.16	1806.80 ± 48.40	37.02 ± 9.74
Cut-off points	24.25	26.71	-	1310.80	29.65

**Table 3 jpm-11-00081-t003:** Association between CEUS quantitative parameters and pathological prognostic factors.

	ER		Ki-67		Her2	
	Negative	Positive	Negative	Positive	Negative	Positive
Peak (%) *	52.6 ± 18.6	56.3 ± 24.1	57.9 ± 23.3	52.1 ± 21.9	53.1 ± 24.4	57.3 ± 21.2
TTP (s) *	14.1 ± 4.5	15.8 ± 8.0	15.7 ± 5.9	15.0 ± 8.7	16.6 ± 8.7	14.2 ± 5.3
MTT (s) **	24.6 (9.2)	21.1 (12.6)	18.7 (49.6)	23.1 (17.4)	20.8 (36.6)	18.1 (49.6)
RBV (mL) **	1436.8 (737.7)	1316.3 (1099.2)	932.3 (3770.4)	2120.5 (1680.9)	908.3 (3046.7)	956.9 (3770.4)
RBF (mL/s) *	56.4 ± 16.6	60.0 ± 25.5	61.1 ± 24.3	56.5 ± 23.8	56.6 ± 25.9	61.4 ± 22.5

* mean ± standard deviation, ** median (range).

**Table 4 jpm-11-00081-t004:** Quantitative parameters variability depending on the tumor’s histological grade.

Tumor Histological Grade	Quantitative Parameters	
	Rise Time (The Time from 10% Maximum Intensity to 90%)	TTP
I or II	9.3 s ± 3.9 s	11.6s ± 6.1 s
III	11.4 s ± 5.4 s	14.7s ± 7.7 s

**Table 5 jpm-11-00081-t005:** Diagnostic performance of CEUS versus the conventional US in differentiating benign and malignant breast lesions—results from a meta-analysis including different studies in the two groups [35].

Pooled Values	CEUS Versus the US		CEUS + US Versus US	
	CEUS	US	CEUS + US	US
Sensitivity	0.93 (95%CI:0.91–0.95)	0.87 (95%CI:0.85–0.90)	0.94 (95%CI:0.92–0.96)	0.87 (95%CI:0.84–0.90)
Specificity	0.86 (95%CI:0.84–0.88)	0.72 (95%CI:0.69–0.75)	0.86 (95%CI:0.82–0.89)	0.80 (95%CI:0.76–0.84)

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
