# Peer review of "Contrast-Enhanced Ultrasonography in the Diagnosis and Treatment Modulation of Breast Cancer"

_jpm, 2021, doi:10.3390/jpm11020081_

Round 1

Reviewer 1 Report

In this manuscript, the authors investigated the method of contrast-enhanced ultrasound in the diagnosis of breast cancer. Breast cancer is very important field with early diagnosis required. CEUS as an imaging technique, can both improve qualitative and quantitative parameters of breast lesions via comparison with other methods. The technique is critical to improve our current understanding of breast cancer diagnosis. The manuscript is well-organized with results support conclusions. The study is suggested to be accepted. 

Author Response

Response to Reviewer 1 Comments

Point 1: In this manuscript, the authors investigated the method of contrast-enhanced ultrasound in the diagnosis of breast cancer. Breast cancer is very important field with early diagnosis required. CEUS as an imaging technique, can both improve qualitative and quantitative parameters of breast lesions via comparison with other methods. The technique is critical to improve our current understanding of breast cancer diagnosis. The manuscript is well-organized with results support conclusions. The study is suggested to be accepted. 

Response 1: Dear reviewer,

We would like to thank you for taking the time to carefully evaluate our article. Thank you for your appreciation and for your positive response.

Best regards,

Ioana Boca

Reviewer 2 Report

Thanks for submitting this interesting work about the role of contrast-enhanced ultrasound in breast cancer in terms of diagnosis, staging and follow-up of the post-treatment response.

The work addresses the problem, the advantages and the limits of the technique well, also through excellent images and a good number of supporting bibliographic references. However, there are some aspects that in my opinion deserve integration to consider a publication on this topic:

  • Lines 160-161: the possibility of predicting some characteristics of the tumor or even the molecular subtype is the subject of some radiomics studies in senology on more techniques that you should mention.As an example you could add this work to your references:Inizio modFine modulo

La Forgia D. et al. Radiomic analysis in contrast-enhanced spectral mammography for predicting breast cancer histological outcome. Diagnostics Volume 10, Issue 9, September 2020, Article number 708

  • Lines 203-204 : I do not understand well how it is possible to compare a two-dimensional technique (CEUS) with a three-dimensional and multiparametric technique (MRI): in the first there is no possibility to evaluate the lymph nodes of the internal mammary chain and also the relationships with the pectoral muscle and the areola-nipple complex are problematic.Can you explain better what was evaluated in this comparison?
  • Lines 238-240 : these limitations reduce the range of lesions assessable with CEUS: did the diagnostic performances described above consider these limits or not?
  • Lines 241-253: among the limits of CEUS there is certainly the presence of macro / microcalcifications which, when grouped in clusters, can be the initial expression of a DCIS or constitute the more or less extensive component of DCIS associated with an invasive carcinoma.This is an important limitation of the CEUS exam that you should add in this paragraph.As a remedy for this type of findings, CAD systems could help improve performance as is already the case in mammography techniques.In this regard, I suggest two bibliographic entries to add to your references:

Fanizzi  A. et al. Ensemble discrete wavelet transform and gray-level co-occurrence matrix for microcalcification cluster classification in digital mammography. Applied Sciences Volume 9, Issue 24, 1 December 2019, Article number 5388

Losurdo L. et al. Radiomics analysis on contrast-enhanced spectral mammography images for breast cancer diagnosis: A pilot study. Entropy Volume 21, Issue 11, 1 November 2019, Article number 1110

  • there is no mention of tumor evaluation in response with percutaneous thermal ablation techniques

Inizio modulo

Fine modulo

Author Response

Response to Reviewer 2 Comments

Point 1: Lines 160-161: the possibility of predicting some characteristics of the tumor or even the molecular subtype is the subject of some radiomics studies in senology on more techniques that you should mention. As an example you could add this work to your references:Inizio modFine modulo

La Forgia D. et al. Radiomic analysis in contrast-enhanced spectral mammography for predicting breast cancer histological outcome. Diagnostics Volume 10, Issue 9, September 2020, Article number 708

Response 1: As a response to Point 1, we added in the body text the following statement (with the suggested citation) -- Lines 165-167:

The possibility of predicting the molecular subtype with the help of CEUS is an important aspect in clinical practice and it is also the subject of innovative radiomic studies in senology [25]. 

Point 2:Lines 203-204 : I do not understand well how it is possible to compare a two-dimensional technique (CEUS) with a three-dimensional and multiparametric technique (MRI): in the first there is no possibility to evaluate the lymph nodes of the internal mammary chain and also the relationships with the pectoral muscle and the areola-nipple complex are problematic.Can you explain better what was evaluated in this comparison?

Response 2: As a response to Point 2, we modified in the body text the title with the following statement – Line 219: “Evaluation of breast lesions: the added value of CEUS to ultrasound and comparison with MRI” as the comparison between the two types of examinations (CEUS and MRI) was only related to the assessment of the enhanced morphological features and dynamic enhancement curves of breast lesions (Lines 241-253)

Point 3:Lines 238-240 : these limitations reduce the range of lesions assessable with CEUS: did the diagnostic performances described above consider these limits or not? 

Response 3:The evaluation of the diagnostic performance of CEUS took into account the mentioned limitations. In some studies were included also young patients (under 35 years), the included lesions were smaller than 20 mm. The depth of the lesion was a criterion that was taken into account only in the study [39].

Point 4:Lines 241-253: among the limits of CEUS there is certainly the presence of macro / microcalcifications which, when grouped in clusters, can be the initial expression of a DCIS or constitute the more or less extensive component of DCIS associated with an invasive carcinoma.This is an important limitation of the CEUS exam that you should add in this paragraph.As a remedy for this type of findings, CAD systems could help improve performance as is already the case in mammography techniques.In this regard, I suggest two bibliographic entries to add to your references:

Fanizzi  A. et al. Ensemble discrete wavelet transform and gray-level co-occurrence matrix for microcalcification cluster classification in digital mammography. Applied Sciences Volume 9, Issue 24, 1 December 2019, Article number 5388

Losurdo L. et al. Radiomics analysis on contrast-enhanced spectral mammography images for breast cancer diagnosis: A pilot study. Entropy Volume 21, Issue 11, 1 November 2019, Article number 1110 

Response 4:As a response to Point 4, we added in the body text the following statement (with the suggested citation) Lines 292-294: To exceed these CEUS limits and improve diagnostic performance, computer-aided design systems could be used as they have been shown to be effective in mammography techniques. [43,44]

Point 5:there is no mention of tumor evaluation in response with percutaneous thermal ablation techniques 

Response 5:As a response to Point 5, we added in the body text the following statements --Lines 268-277:

CEUS is a useful imaging method in assessing the success rate of microwave ablation (MVA) for benign breast lesions. Intralesional and perilesional enhancement assessment before and one hour after MVA at CEUS and MRI are useful in assessing the effectiveness of ablation, with a success rate of 87.32% for CEUS and 82.93% for MRI [40].

Yuanzhi et al. [41] went further in evaluating the usefulness of CEUS and SonoVue®. The authors analyzed in "in vivo" and "ex vivo" animal models studies the improvement of the therapeutic efficacy of high intensity focused ultrasound (HIFU) thermal ablation of breast tumors by using SonoVue® microbubbles. The microbubbles present at the level of the tumor act as cavitation nuclei and enhance tissue heatening during HIFU treatment producing coagulation necrosis.

Reviewer 3 Report

In the manuscript

Contrast-enhanced ultrasonography in the diagnosis and treatment modulation of breast cancer”,

the authors give a very nice discussion and review about the current state of CEUS in breast cancer and its impact on treatment. The paper is well written and structured very well. Also the figures and example images are very nice. I would just recommend to carefully check for typos and grammar errors. Otherwise I have just these few comments:

  • The tables need to be adapted to the size of the text.
  • Section 5 (“Usefulness of CEUS”) is way to short, please extend it, there is plenty information which can be presented here.

Author Response

Response to Reviewer 3 Comments 

Point 1:the authors give a very nice discussion and review about the current state of CEUS in breast cancer and its impact on treatment. The paper is well written and structured very well. Also the figures and example images are very nice. I would just recommend to carefully check for typos and grammar errors. Otherwise I have just these few comments:

Response 1:The manuscript was once again checked for spelling and grammar errors. 

Point 2:The tables need to be adapted to the size of the text.

Response 2:All tables have been modified according to your suggestion.

Point 3:Section 5 (“Usefulness of CEUS”) is way to short, please extend it, there is plenty information which can be presented here.

Response 3:

As a response to Point 3, we added in the body text the following statement--Lines 250-277:

In addition to the above mentioned benefits or indications of CEUS: the possibility of differentiating between benign and malignant lesions, the possibility of predicting the molecular subtype, detection and characterization of LCs, SLN and predicting the response to neoadjuvant chemotherapy, CEUS has some notable advantages. Compared to MRI, CEUS is an accessible imaging technique, the examination time is shorter, the patient are examined in a comfortable position and it can be performed on patients with ferromagnetic metal implants, pacemakers, claustrophobia or with contraindication to Gadolinium administration.

CEUS is a useful imaging method in assessing the success rate of microwave ablation (MVA) for benign breast lesions. Intralesional and perilesional enhancement assessment before and one hour after MVA at CEUS and MRI are useful in assessing the effectiveness of ablation, with a success rate of 87.32% for CEUS and 82.93% for MRI [40].

Yuanzhi et al. [41] went further in evaluating the usefulness of CEUS and SonoVue®. The authors analyzed in "in vivo" and "ex vivo" animal models studies the improvement of the therapeutic efficacy of high intensity focused ultrasound (HIFU) thermal ablation of breast tumors by using SonoVue® microbubbles. The microbubbles present at the level of the tumor act as cavitation nuclei and enhance tissue heatening during HIFU treatment producing coagulation necrosis.

Round 2

Reviewer 2 Report

I thank the authors for replying to my comments. In my opinion the manuscript is now ready for publication. Congratulations!